# A retrospective cohort study to observe the efficacy and safety of Endoscopic Submucosal Dissection (ESD) with adjuvant radiotherapy for T1a-MM/T1b-SM Esophageal Squamous Cell Carcinoma (ESCC)

**Hongna Lu[1], Yanping Bei[2], Chunnian Wang[3], Xi Deng[3], QinQin Hu[1], Wenying Guo[1], Xuesong Zhang[4] \***

**1** Department of Gastroenterology, Ningbo Medical Center Lihuili Hospital, Ningbo, Zhejiang, China, **2** Department of Radiotherapy, Ningbo Medical Center Lihuili Hospital, Ningbo, Zhejiang, China, **3** Ningbo Clinical and Pathological Diagnosis Center, Ningbo, Zhejiang, China, **4** Endoscopy Center, Sir Run Run Shaw Hospital Affiliated to Zhejiang University School of Medicine, Hangzhou, Zhejiang, China

\* 21316838@qq.com

## Abstract

### Background and aim

The clinical outcome of endoscopy submucosal dissection with subsequent radiotherapy for esophageal squamous cell carcinoma remain unclear. In this study we aim to investigate the efficacy and safety of endoscopic submucosal dissection with adjuvant radiotherapy in the treatment of superficial esophageal squamous cell carcinoma involving the muscularis mucosae (T1a-MM) or the submucosa < 200 μm (T1b-SM1).

### Methods

We analyzed 20 patients with pathologically confirmed T1a-MM or T1b-SM1 esophageal squamous cell carcinoma treated by endoscopic submucosal dissection from 2016 to 2020 in Lihuili Hospital, 9 patients received adjuvant radiotherapy (RT group) and 11 patients received did not (non-RT group).

### Results

All 20 patients underwent en bloc resection, and both the vertical and horizontal margins were negative. There was no recurrence or lymph node metastasis in the RT group, and no serious complications or death were observed. In the non-RT group, 2 patients had local recurrence and 1 had distant metastasis. None of the 20 patients died of esophageal carcinoma.

**Data Availability Statement:** All raw data cannot be shared publicly because of personal data

protection and privacy of the patients. Data are available from the Ningbo Medical Center Lihuili Hospital Ethics Committee (email: lihuiliethics@163.com) for researchers who meet the apply criteria for access to confidential data.

**Funding:** The authors received no specific funding for this work.

**Competing interests:** The authors have declared that no competing interests exist.

## Conclusions

Adjuvant radiotherapy following endoscopic submucosal dissection may be a safe and effective method for the treatment of T1a-MM/T1b-SM1 superficial esophageal squamous cell carcinoma.

## 1. Introduction

The treatment options for superficial esophageal squamous cell carcinoma (ESCC) depend on the risk of lymph node metastasis [1]. Patients with pathologically confirmed T1a-EP (intraepithelial carcinoma) or pT1a-LPM (carcinoma confined within the lamina propria) ESCC have a low risk of lymph node metastasis, thus endoscopic submucosal dissection (ESD) is considered for radical tumor resection, and adjuvant treatment is not deemed necessary [2, 3]. However, patients with superficial ESCC involving the muscularis mucosae (T1a-MM) or the submucosa < 200 μm (T1b-SM1) are at an elevated risk of metastasis, necessitating the consideration of adjuvant treatment.

Specimens from pT1a-MM ESCC patients (inclusion of vascular invasion) who underwent surgical resection as the first-line treatment had concurrent lymph node metastasis in 0–26.7% of the cases, and further analysis showed that 29 out of the 199 patients had concurrent lymph node metastasis (14.6%, 95% CI: 10.0–20.3%) [4]. Eguchi T et. al analyzed 50 pT1a-MM ESCC patients and discovered that patients with vascular invasion have a higher risk of lymph node metastasis [negative vascular invasion 4/38 (10.5%) vs positive vascular invasion 5/12 (41.7%)] [2]. Since the incidence of lymph node metastasis depends on whether the blood vessels are infiltrated, the Esophageal cancer practice guidelines 2017 edited by the Japan Esophageal Society [5] suggested that: "It is strongly suggested to have adjuvant treatment for T1a-MM ESCC patients with vascular invasion".

The incidence of lymph node metastasis after surgical resection in T1b-SM1 ESCC patients (inclusion of vascular invasion) ranged from 8.3–53.1% summarized from previous studies [2, 6–12]. The 2020 Endoscopic submucosal dissection/endoscopic mucosal resection guidelines for esophageal cancer showed that after surgical resection, 43 out of 170 T1b-SM1 ESCC patients had lymph node metastasis (25.3%, 95% CI. 19.0%-32.5%) [4]. Therefore, the guidelines for endoscopic submucosal dissection/endoscopic mucosal resection of esophageal cancer released in 2020 [13] state, "For patients with pT1b-SM esophageal squamous cell carcinoma after endoscopic resection, it is strongly recommended to consider additional surgical resection or adjuvant chemoradiotherapy." The document also notes that in the follow-up observation group, the metastasis rate for T1a-MM esophageal squamous cell carcinoma patients after endoscopic resection without vascular invasion is 5.6%. However, considering the potential reduction in quality of life and the risk of treatment-related mortality due to additional surgical resection, as well as the delayed adverse events and treatment-related mortality associated with additional chemoradiotherapy, the guidance committee did not conclude whether to recommend additional treatment. Regarding the increased incidence of lymph node metastasis in pT1a-MM/ pT1b-SM1 ESCC patients, the committee agreed that adjuvant treatment is necessary after ESD to prevent lymph node metastasis [13].

Several studies have reported adjuvant chemotherapy for pT1a-MM/pT1b-SM1 ESCC patients after ESD [14–19], but only a few studies have performed adjuvant radiotherapy [20–22]. Therefore, in this study, we retrospectively analyzed the safety and efficacy of ESD combined with adjuvant radiotherapy for T1a-MM/T1b-SM1 ESCC patients.

## 2. Methods

### 2.1 Subjects demographics

This retrospective, observational, cohort study screened 160 ESCC patients underwent ESD treatment at Ningbo Medical Center Lihuili Hospital between 2016 to 2020, as shown in **Fig 1**. The vertical and horizontal margins were all negative after surgical resection. Among them, 24 patients were pathologically confirmed to be pT1a-MM or pT1b-SM1 cases after ESD treatment. Complementary radiotherapy was performed on 9 cases consisting of 5 T1a-MM patients and 4 T1b-SM1 patients. One T1b-SM1 subject was pathologically diagnosed to have lymphatic invasions. Out of the 24 patients, 11 had only follow-up but did not undergo adjuvant treatment, 7 were T1a-MM and 4 were T1b-SM1. Besides the patients receiving adjuvant radiotherapy or did not undergo adjuvant treatment, the remaining 4 ESCC patients underwent adjuvant surgical esophagectomy, and postoperative esophageal specimens were free of tumor residue and metastatic lymph nodes and was excluded. Therefore, 20 ESCC patients met the inclusion criteria for this study, except for the 4 pT1b-SM1 patients who underwent adjuvant surgery. No lymphatic metastasis was observed by contrast-enhanced CT (CECT) scans from the cervical to the abdominal region before treatment.

The data was obtained from the Ningbo Medical Center Lihuili Hospital and the data collection period was from January 1, 2016 through December 31, 2020. Due to the COVID-19 pandemic from 2021 to 2022, the medical center and medical utilization were used priority for patients with COVID-19 positive, so we didn't apply for the data during the COVID-19 pandemic. Furthermore, the most important is that this study was approved by the Ethics Committee of Biomedical Research Involving Humans of Ningbo Medical Center Lihuili Hospital on March 16, 2023 (IRB number: KY2023SL054-01). And we received the data from a hospital-based release on March 30, 2023 after the IRB approval without any human rights concerns.

All raw data cannot be shared publicly because of personal data protection and privacy of the patients. Data are available from the Ningbo Medical Center Lihuili Hospital Ethics Committee for researchers who meet the apply criteria for access to confidential data. Before the surgery, all patients provided their consent by signing a written informed consent form in the local language, which was approved by the Ethics Committee.

### 2.2 Pathology evaluation

Specimens were evaluated pathologically according to the Japan Esophageal Society guidelines: pT1a-MM is cancer cells infiltrating the mucosal muscle, and pT1b-SM1 is cancer cells infiltrating the superficial submucosal layer ($< 200$ μm). All specimens were pathologically evaluated for tissue type, depth of infiltration, vertical margin, horizontal margin, and the presence or absence of vascular invasion.

### 2.3 Radio therapy

Radiotherapy was administered using intensity-modulated radiation therapy (IMRT) within one month after ESD. A titanium clip was endoscopically placed in the tumor bed (post-ESD scar) before the localization CT to indicate the upper and lower borders. The GTVtb defined as the upper and lower esophageal border was determined by these titanium clips, and the PGTVtb was the 1 cm outward expansion of GTVtb.

The lymph node prophylactic irradiation area was determined based on the specific scope of the esophageal segment. For the cervical and upper thoracic esophageal segment, it included the supraclavicular lymphatic drainage area, paraesophageal area, and stations 2, 4, 5, and 7.

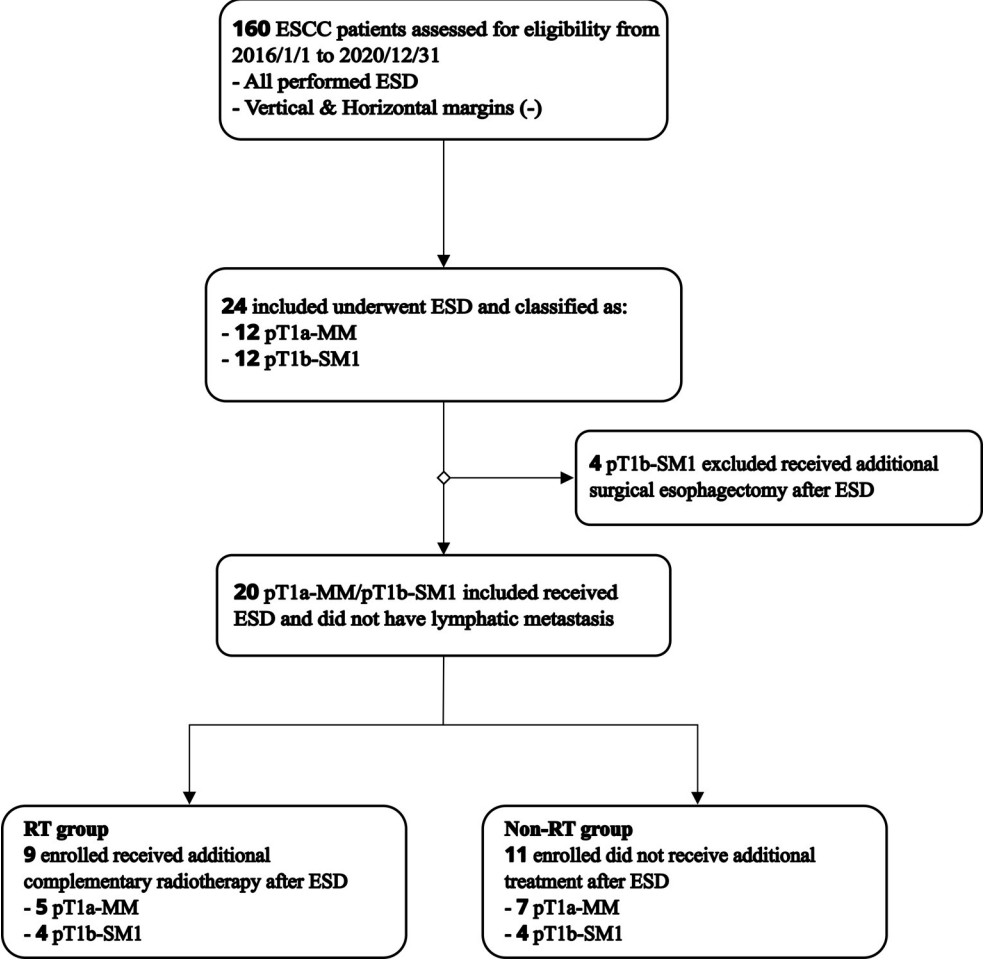

**Fig 1. Flow chart of study population.**

The middle segment included the paraesophageal area and stations 2, 4, 5, and 7. The lower segment included the paraesophageal area, stations 4, 5, and 7, and the lymphatic drainage area around the left side of the stomach and cardiac. The regional lymph node stations were divided based on the American AJCC-UICC 1996 grouping of chest lymph nodes [23].

The lungs, spinal cord, and heart in particular the left anterior descending artery were considered organs at risk during radiation therapy. In addition, many articles suggest that the radiation dose to the Left Anterior Descending (LAD) artery of the heart is closely related to cardiac functional impairment and prognosis. Therefore, in our center, we consider the LAD as an organ at risk, rigorously restricting the radiation dose to a limit of mean less than 25 Gy and V40 less than 20 [24, 25]. The prescribed dose for the Planning Target Volume (PTV) is 41.4 Gy in 23 fractions, and the dose for the Planning Gross Tumor Volume of the primary tumor bed (PGTVtb) is 50.4 Gy in 28 fractions. [14, 17, 18].

Treatment-related toxicities were scored according to the Common Terminology Criteria for Adverse Events, version 4.0. Acute toxicity was assessed weekly during radiation, and late toxicity was defined as events that occurred or persisted more than 2 months after completion of radiation [26].

## 2.4 Follow-up

Gastroscopy was performed to detect any local recurrence every 3–6 months within the first year after surgery. CT scans were also carried out to observe lymph nodes and distal metastasis in the cervical, thoracic, and abdominal region every 6 months during the first year after gastroscopy, and then annually afterwards.

## 2.5 Statistical methods

The statistical analysis in this study was processed by the analysis software Prism 5.0. T-test and $\chi^2$ test were used to analyze the correlation between factors such as gender, age, number and location of lesions, and follow-up results, such as tumor recurrence and metastasis. The progression-free survival (PFS) was evaluated by Kaplan–Meier analysis. A p-value of less than 0.05 using two-sided tests indicated statistical significance.

# 3. Results

The 20 patients who underwent ESD surgery were pathologically evaluated to be pT1a-MM or pT1b-SM1 types. The tumor was fully resected and the vertical margins and the horizontal margins were negative. We separate the 20 patients based on whether they received adjuvant radiotherapy (RT group, N = 9) or not (non-RT group, N = 11) to evaluate the safety and efficacy of the adjuvant treatment.

Patient characteristics of the 9 subjects who received adjuvant radiotherapy defined as the RT group are summarized in Table 1. There were 7 males and 2 females, with an average age of 66.1 years. The tumor locations of the upper, middle, and lower (U:M:L) esophagus were 1:3:5 among the patients, with an average tumor size of 28.6 mm (15–51 mm). Out of the 9 cases, 5 were T1a-MM and 4 were T1b-SM1. Of the T1b-SM1 patients, 1 was pathologically confirmed with lymphatic invasion after ESD. The patient had an ESD resection area of > 3/4 circumference, and esophageal stricture was developed. Esophageal balloon dilation was performed 6 times to relieve dysphagia. There were no postoperative complications, such as bleeding, perforation, or infection of the patient. For the complications that occurred in the RT group of patients, 2 had chest allodynia, and 1 had grade 1 radiation pneumonitis, but all recovered within 2 weeks after completion of radiotherapy. There was no grade 3 radiation-related complications, such as severe radiation pneumonitis, esophagitis, bone marrow suppression, severe radiation dermatitis, and fistula formation. The mean follow-up period was

**Table 1. Clinical characteristics of ESCC patients who underwent ESD with adjuvant radiotherapy group (RT Group).**

| Case | Sex | Age (years) | Location | Size (mm) | Circumference | Invasion depth | Horizontal margin | Vertical margin | Vascular infiltration | Follow-up period (months) |
|------|--------|-------------|----------|-----------|---------------|----------------|-------------------|-----------------|-----------------------|---------------------------|
| 1 | Male | 65 | L | 15*15 | 1/5 | MM | - | - | - | 15 |
| 2 | Male | 63 | L | 25*22 | 1/3 | MM | - | - | - | 45 |
| 3 | Male | 72 | M | 34*25 | 1/2 | SM1 | - | - | - | 25 |
| 4 | Male | 68 | U | 24*20 | 1/3 | MM | - | - | - | 38 |
| 5 | Male | 54 | M | 36*32 | 3/4 | SM1 | - | - | + | 18 |
| 6 | Male | 66 | L | 51*40 | 5/6 | SM1 | - | - | - | 20 |
| 7 | Male | 67 | M | 30*28 | 2/3 | SM1 | - | - | - | 8 |
| 8 | Female | 71 | L | 18*15 | 1/5 | MM | - | - | - | 25 |
| 9 | Female | 69 | L | 25*20 | 1/3 | MM | - | - | - | 36 |

U: upper esophagus M: middle esophagus, L: lower esophagus, MM: muscularis mucosae, SM1: submucosa < 200μm

25.6 months (8–45 months), with no death, no recurrence, or lymph node metastasis during the follow-up period.

Patient characteristics of the 11 patients in the non-RT group are presented in **Table 2**. There were 8 males and 3 females, with an average age of 60.7 years. The tumor locations of the U:M:L esophagus were 2:4:5 in the population, with an average tumor size of 27.4 mm (10–50 mm). Among them, 7 were T1a-MM and 4 were T1b-SM1, with no vascular invasion. There was 1 patient with a resection area of > 3/4 circumference who had developed esophageal stricture. Esophageal balloon dilation was performed 10 times to relieve dysphagia. The patient had no postoperative complications, such as bleeding, perforation, or infection.

We compared the clinicopathological characteristics of the RT group and non-RT group of patients after the ESD surgery and listed them in **Table 3**. As for the disease progression features, no patients in the RT group exhibited recurrence, metastasis, or death compared to 4 in the non-RT group. During the follow-up period for the non-RT group, 2 patients had a regional recurrence of ESCC, and 1 T1a-MM patient had abdominal lymph node metastases after 8 months of follow-up. Although no patients died from ESCC, 1 died from laryngeal cancer during the follow-up period in the non-RT group. The non-recurrence rate of the RT group compared with the non-RT group is 100% to 81.8% (p = 0.18). In summary, there was no difference between the clinicopathological characteristics of the 2 cohorts.

To validate the safety and efficacy of the adjuvant radiotherapy after ESD for ESCC compared with ESD only, progression-free survival (PFS) was assessed and summarized in **Table 4**. We defined PFS as since enrollment, the disease progression, metastasis, or death from any cause, whichever occurred first. There were 0 death (0.0%) out of 9 patients in the RT group while there were 4 deaths (36.4%) out of 11 patients in the non-RT group. As shown in **Fig 2,** no progression events occurred in the RT group leading to a better PFS probability compared to the non-RT group, implying a progression free trend for the usage of adjuvant radiotherapy (p = 0.076). The median PFS time was not reached by either the RT or non-RT groups.

## 4. Discussion

The current definition of "early esophageal cancer" is intramucosal carcinoma (T1a) with or without lymph node metastasis [27], which is further classified into subcategories according to

**Table 2. Clinical characteristics of ESCC patients who underwent ESD only group (Non-RT Group).**

| Case | Sex | Age (years) | Location | Size (mm) | Circumference | Invasion depth | Horizontal margin | Vertical margin | Vascular infiltration | Follow-up period (months) |
|---|---|---|---|---|---|---|---|---|---|---|
| 1 | Male | 60 | L | 10*10 | 1/6 | MM | - | - | - | 27 |
| 2 | Male | 58 | L | 28*20 | 1/3 | SM1 | - | - | - | 25 |
| 3 | Male | 55 | M | 20*15 | 1/4 | MM | - | - | - | 25 |
| 4 | Male | 63 | U | 30*20 | 1/3 | MM | - | - | - | 40 |
| 5 | Male | 62 | M | 35*35 | 3/4 | SM1 | - | - | - | 12 |
| 6 | Male | 57 | L | 50*20 | 1/4 | MM | - | - | - | 15 |
| 7 | Male | 60 | M | 25*15 | 1/4 | SM1 | - | - | - | 9 |
| 8 | Male | 80 | L | 30*30 | 2/3 | MM | - | - | - | 20 |
| 9 | Female | 66 | L | 25*20 | 1/3 | SM1 | - | - | - | 18 |
| 10 | Female | 48 | U | 23*15 | 1/4 | MM | - | - | - | 36 |
| 11 | Female | 59 | M | 25*22 | 1/3 | MM | - | - | - | 46 |

U: upper esophagus M: middle esophagus, L: lower esophagus, MM: muscularis mucosae, SM1: submucosa < 200μm

**Table 3. Clinicopathological characteristics of ESCC patients after ESD with or without radiotherapy.**

| Characteristics | | RT group (N = 9) | Non-RT group (N = 11) | P value |
|---|---|---|---|---|
| Average age (years) | | 66.1 | 60.7 | 0.099 |
| Sex | Male | 7 | 8 | 0.795 |
| | Female | 2 | 3 | |
| Number of synchronous | Solitary | 8 | 9 | 0.660 |
| | Multiple | 1 | 2 | |
| Location | U | 1 | 2 | 0.870 |
| | M | 3 | 4 | |
| | L | 5 | 5 | |
| Invasion depth | pT1a-MM | 5 | 7 | 0.714 |
| | pT1b-SM1 | 4 | 4 | |
| Average size (mm) | | 28.7 | 27.4 | 0.782 |
| Circumference | < 3/4 | 7 | 10 | 0.413 |
| | ≥ 3/4 | 2 | 1 | |
| Horizontal margin (+:−) | | 0:9 | 0:11 | 1 |
| Vertical margin (+:−) | | 0:9 | 0:11 | 1 |
| Vascular infiltration (+:−) | | 1:8 | 0:11 | 0.257 |
| Average follow-up period (months) | | 25.6 | 24.8 | 0.892 |
| Recurrence | | 0 | 2 | 0.176 |
| Lymph node or distal metastasis | | 0 | 1 | 0.353 |
| Death | | 0 | 1 | 0.353 |

U: upper esophagus M: middle esophagus, L: lower esophagus, MM: muscularis mucosae, SM1: submucosa < 200μm

the depth of infiltration: carcinoma in situ (T1a-EP), lamina propria mucosae (T1a-LPM), and mucosal muscle (T1a-MM) [28] carcinoma. The current definition of "superficial esophageal cancer" is intramucosal (T1a) and submucosal (T1b), with or without lymph nodes or distant organ metastasis.

**Table 4. Summary of progression free survival.**

| Characteristics | RT Group (N = 9) | Non-RT Group (N = 11) | Total (N = 20) |
|---|---|---|---|
| **Progression free survival** | | | |
| Event, n (%) | 0 (0.0%) | 4 (36.4%) | 4 (20.0%) |
| Censor, n (%) | 9 (100.0%) | 7 (63.6%) | 16 (80.0%) |
| **Median survival period (Months)** | | | |
| Median Survival Time | --- | --- | --- |
| (Q1, Q3) | (---, ---) | (25.0, ---) | (25.0, ---) |
| **Follow-up period (Months)** | | | |
| Number | 9 | 11 | 20 |
| Mean (STD) | 25.3 (12.0) | 24.8 (11.8) | 25.1 (11.6) |
| Median | 23.0 | 25.0 | 24.0 |
| Range | (8.0, 45.0) | (9.0, 46.0) | (8.0, 46.0) |
| 95% C.I. | (16.09, 34.57) | (16.90, 32.74) | (19.63, 30.47) |

Progression-free survival (PFS) is defined as the time from enrollment to disease progression, metastasis, or death from any cause, whichever occurred first

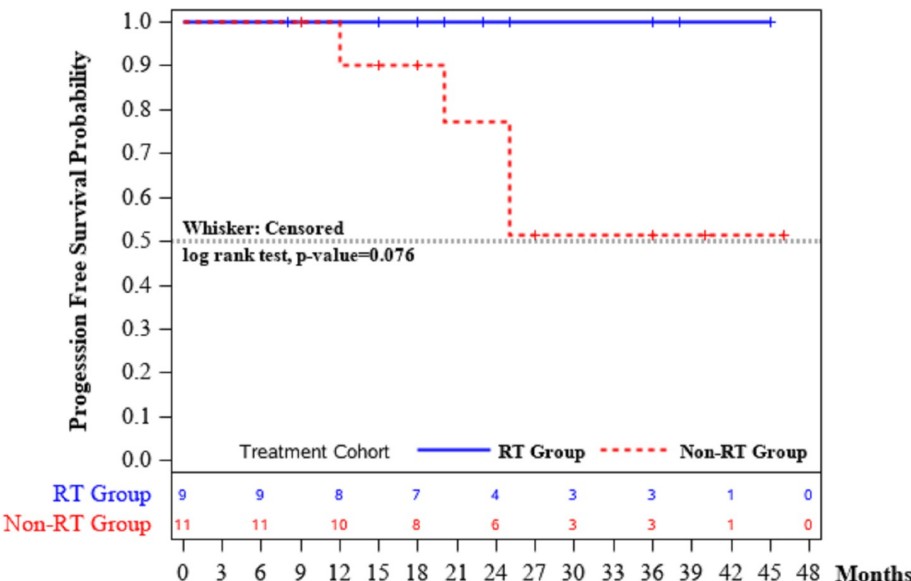

**Fig 2. Survival curve.** Each group is displayed in a different color.

ESD is currently the most effective and relatively safe treatment of esophageal cancer. However, ESD is not a panacea, and recurrence is sometimes inevitable in patients with early-stage ESCC. The 3-year recurrence-free survival (RFS) of ESD for stage T1 ESCC ranges from 57.8% to 91.5% according to the literature [20, 29, 30]. Considering the enhancing lymph node metastasis incidence of patients with pT1a-MM/pT1b-SM1 ESCC, adjuvant treatment is required after ESD to prevent lymph node metastasis [13]. The results of this study demonstrated that combined adjuvant radiotherapy after ESD is a safe and effective way to treat pT1a-MM/pT1b-SM1 ESCC patients, which reduced the tumor recurrence and metastasis rate.

In 2011, a retrospective study reported 14 patients with ESCC (8 T1a-MM, 4 T1b-SM1, and 2 T1b-SM2) underwent combined chemoradiotherapy within 2 to 4 weeks after ESD. Post-ESD pathology suggested negative margins and vascular infiltration in all specimens, while 2 cases had lymph node involvement. After a follow-up period of up to 19 to 70 months, none of the patients died of esophageal cancer, and no local recurrence, lymph node, or distant metastasis was detected in all but 2 died of other diseases [15]. Kawaguchi et al. also compared the effectiveness of ESD combined with chemoradiotherapy (ESD-CRT) versus chemoradiotherapy alone (dCRT) in treating T1a-MM/T1b-SM ESCC patients in a retrospective study in 2015. The 3-year overall survival rates were 63.2% and 90.0% in the dCRT and ESD-CRT groups, respectively. [16] In the dCRT group, 9 patients (29.0%) had tumor recurrence, whereas only 1 patient (6.3%) in the ESD-CRT group had a recurrence. While in the dCRT group, 3 patients died from primary cancer, 3 from treatment-related adverse events, and 6 from other causes. In comparison, none of the ESD-CRT group died from primary disease and 2 died from myocardial infarction [16]. Hisano et al. retrospectively analyzed 27 pathologically confirmed T1a-MM/T1b-SM1 ESCC patients after ESD in 2018, 13 patients (6 T1a-MM and 7 T1b-SM1) received adjuvant radiotherapy after ESD (RT group) and the remaining 14 patients did not receive adjuvant radiotherapy after ESD (non-RT group). The 3-year locoregional control in the RT and the non-RT group is 100% and 57.8% (p = 0.02), and the 3-year cause-specific survival is 82.1% and 77.8% (p = 0.80), respectively [20].

Recently, Zhang et al. conducted a prospective randomized controlled trial to validate the effectiveness of combining ESD with adjuvant radiotherapy to control early ESCC (T1a-EP/

LPM/MM) recurrence. They randomized 70 patients with pathologically confirmed stage T1a ESCC treated with ESD, 35 of whom received 40–66.4 Gy radiation within 2 months after ESD. The non-RT group had 3 cases of tumor recurrence and none in the radiotherapy group. There were no local lymph nodes or distant metastasis, or death in both groups. The 3-year cumulative recurrence-free survival rate was 100% in the adjuvant radiotherapy group and 85.3% in the non-radiotherapy group, and there were also no serious radiation-related complications in the radiotherapy group [22].

In this study, the combination of complete resection of the lesion by ESD with adjuvant radiotherapy to control local recurrence and metastasis resulted in better recurrence-free and long-term survival rates for patients with T1a-MM/T1b-SM1 ESCC. Although the difference in recurrence-free survival between patients with esophageal cancer in the RT group and the non-RT group was not statistically significant, none of the patients in the RT group developed local recurrence or distant metastasis, and none of the patients died from a recurrence of esophageal cancer or the complications caused by it. The findings suggest that the addition of adjuvant radiotherapy is both safe and effective, offering promising results in the overall treatment strategy for this patient population. However, the study was a single-center retrospective study with a small number of cases. Future studies should include more subjects with longer follow-up periods and perform case-control studies to better exhibit the effectiveness and safety of ESD in combination with radiotherapy for patients with T1a-MM/T1b-SM1 esophageal squamous cell carcinoma.

## Acknowledgments

We would like to acknowledge Formosa Biomedical Technology Corporation for its participation in the data analysis, writing, and editing of the manuscript.

## Author Contributions

**Conceptualization:** Yanping Bei, Xuesong Zhang.

**Data curation:** Hongna Lu, Xuesong Zhang.

**Formal analysis:** Hongna Lu, Chunnian Wang, Xuesong Zhang.

**Funding acquisition:** Chunnian Wang.

**Investigation:** Chunnian Wang.

**Methodology:** Yanping Bei, Wenying Guo.

**Project administration:** Xi Deng.

**Resources:** Xi Deng, Wenying Guo.

**Software:** Xi Deng.

**Supervision:** QinQin Hu.

**Validation:** QinQin Hu, Wenying Guo.

**Visualization:** Hongna Lu.

**Writing – original draft:** Hongna Lu, Xuesong Zhang.

**Writing – review & editing:** Hongna Lu, Yanping Bei, Xuesong Zhang.

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
