## [Decision Letter · Decision Letter 0]

11 Dec 2023

PONE-D-23-34928A Retrospective Cohort Study to Observe the Efficacy and Safety of Endoscopic Submucosal Dissection (ESD) with Additional Radiotherapy for T1a-MM/T1b-SM Esophageal Squamous Cell Carcinoma(ESCC)PLOS ONE

Dear Dr. Xuesong,

Thank you for submitting your manuscript to PLOS ONE. After careful consideration, we feel that it has merit but does not fully meet PLOS ONE’s publication criteria as it currently stands. Therefore, we invite you to submit a revised version of the manuscript that addresses the points raised during the review process.

We look forward to receiving your revised manuscript.

Kind regards,

Paolo Aurello

Academic Editor

PLOS ONE

Journal Requirements:

4. One of the noted authors is a group or consortium Department of Gastroenterology, Ningbo Medical Center Lihuili Hospital,Ningbo Clinical and Pathological Diagnosis Center, Zhejiang. In addition to naming the author group, please list the individual authors and affiliations within this group in the acknowledgments section of your manuscript. Please also indicate clearly a lead author for this group along with a contact email address.

Reviewers' comments:

Reviewer's Responses to Questions

**Comments to the Author**

1. Is the manuscript technically sound, and do the data support the conclusions?

Reviewer #1: Yes

Reviewer #2: Yes

2. Has the statistical analysis been performed appropriately and rigorously? 

Reviewer #1: Yes

Reviewer #2: Yes

3. Have the authors made all data underlying the findings in their manuscript fully available?

Reviewer #1: No

Reviewer #2: No

4. Is the manuscript presented in an intelligible fashion and written in standard English?

Reviewer #1: Yes

Reviewer #2: Yes

5. Review Comments to the Author

Reviewer #1: Though it is a retrospective study, still some questions are there to find out.

1. It will remain a question to authors why radiotherapy is used in case of T1a-MM without LVSI.

2. What is craniocaudal margin used in radiotherapy portal.

3. Grades of Radiotherapy toxicity (Grade 1,2,3) data will be interesting to know.

Reviewer #2: 1. Its better to replace word "additional" with "Adjuvant" in Title and elsewhere

2. Abstract- line 25, replace received with underwent.

3. Introduction

a. Rewrite 45-48 line as your data do not match with reference

b. Reference 10, may not exactly show LN mets incidence cause its done as salvage so may not be appropriate reference here

c. Rewrite sentence 61-63

d. In Radiotherapy section, in line 128, was Left Anterior descending artery contour as OAR? It is considered most important OAR in heart.

e. Dose of Radiotherapy have differed significantly (41.4 Gy vs 50.4Gy). how was dose decided?

f. In follow up section, How frequently was CT scan done?

g. In result section, complication should be reported based on grade rather than using word like mild.

h. In discussion section, since it is retrospective study of small sample rather than comparing RT vs Non RT group, it is better for author to conclude that adjuvant RT is safe and effective with good results

6. PLOS authors have the option to publish the peer review history of their article (what does this mean?). If published, this will include your full peer review and any attached files.

Reviewer #1: **Yes: **ABHIJIT DAS MD. DNB

Reviewer #2: **Yes: **Dr Simit SApkota

---

## [Author Response · Author response to Decision Letter 0]

23 Jan 2024

Reviewers' comments:

Reviewer's Responses to Questions

Comments to the Author

1. Is the manuscript technically sound, and do the data support the conclusions?

Reviewer #1: Yes

Reviewer #2: Yes

2. Has the statistical analysis been performed appropriately and rigorously? 

Reviewer #1: Yes

Reviewer #2: Yes

3. Have the authors made all data underlying the findings in their manuscript fully available?

Reviewer #1: No

Reviewer #2: No

Reply:

Due to the nature of this clinical study and privacy considerations, the provision of personal information is restricted.

4. Is the manuscript presented in an intelligible fashion and written in standard English?

Reviewer #1: Yes

Reviewer #2: Yes

5. Review Comments to the Author

Reviewer #1: Though it is a retrospective study, still some questions are there to find out.

1. It will remain a question to authors why radiotherapy is used in case of T1a-MM without LVSI.

Reply:

Thank you for the feedback. 

In the background, it has been mentioned that for patients with T1a-MM esophageal squamous cell carcinoma without vascular invasion after endoscopic resection, the metastasis rate in the follow-up observation group is 5.6%. However, considering that additional surgical resection may result in a decreased quality of life and potential treatment-related mortality, as well as the delayed adverse events and treatment-related mortality associated with additional chemoradiotherapy, the guidance committee did not reach a conclusion on whether to recommend additional treatment.

2. What is craniocaudal margin used in radiotherapy portal.

Reply:

The margin is described in the radiotherapy section: “A titanium clip was endoscopically placed in the tumor bed (post-ESD scar) before the localization CT to indicate the upper and lower borders. The GTVtb defined as the upper and lower esophageal border was determined by these titanium clips, and the PGTVtb was the 1 cm outward expansion of GTVtb.”

3. Grades of Radiotherapy toxicity (Grade 1,2,3) data will be interesting to know.

Reply:

Thank you for the feedback, we revised the term according to the criteria listed in the Common Terminology Criteria for Adverse Events, version 4.0. 

Reviewer #2: 

1. Its better to replace word "additional" with "Adjuvant" in Title and elsewhere

Reply:

Thank you for your feedback. We have made revisions accordingly.

2. Abstract- line 25, replace received with underwent.

Reply:

Thank you for your feedback. We have made revisions accordingly.

3. Introduction

a. Rewrite 45-48 line as your data do not match with reference

Reply:

Thank you for your observation. The data used in lines 45-48 is derived from reference 4, and we have modified it and changed the citation. We'll be happy to revisit and make adjustments as needed. If you have any specific concerns or suggestions, please feel free to share.

(4) Ishihara R, Arima M, Iizuka T, Oyama T, Katada C, Kato M, et al. Endoscopic submucosal dissection/endoscopic mucosal resection guidelines for esophageal cancer. Dig Endosc. 2020 May;32(4):452–93.

b. Reference 10, may not exactly show LN mets incidence cause its done as salvage so may not be appropriate reference here

Reply:

Thank you for the feedback. Upon review, we agree that this reference does not suit this context and, as a result, it has been removed.

c. Rewrite sentence 61-63

Reply:

Thank you for your feedback. We have made revisions accordingly to have a clearer view of the background and the guidelines.

d. In Radiotherapy section, in line 128, was Left Anterior descending artery contour as OAR? It is considered most important OAR in heart.

Reply:

Thank you for your feedback. We consider Left Anterior descending artery as OAR, and have made revisions accordingly.

e. Dose of Radiotherapy have differed significantly (41.4 Gy vs 50.4Gy). how was dose decided?

Reply:

The radiotherapy doses have been determined based on historical references, and we have incorporated this information in the methods section.

f. In follow up section, How frequently was CT scan done?

Reply:

Thank you for the feedback, CT was done every 6 months during the first year after gastroscopy, and then annually afterward.

g. In result section, complication should be reported based on grade rather than using word like mild.

Reply:

Thank you for the feedback, we revised the term according to the criteria listed in the Common Terminology Criteria for Adverse Events, version 4.0. 

h. In discussion section, since it is retrospective study of small sample rather than comparing RT vs Non RT group, it is better for author to conclude that adjuvant RT is safe and effective with good results

Reply:

Thank you for the feedback, we have made revisions accordingly.

---

## [Editor Report · Decision Letter 1]

31 Jan 2024

A Retrospective Cohort Study to Observe the Efficacy and Safety of Endoscopic Submucosal Dissection (ESD) with Adjuvant Radiotherapy for T1a-MM/T1b-SM Esophageal Squamous Cell Carcinoma (ESCC)

PONE-D-23-34928R1

Dear Dr. Xuesong Zhang

We’re pleased to inform you that your manuscript has been judged scientifically suitable for publication and will be formally accepted for publication once it meets all outstanding technical requirements.

Kind regards,

Paolo Aurello

Academic Editor

PLOS ONE

---

## [Editor Report · Acceptance letter]

13 Feb 2024

PONE-D-23-34928R1 

PLOS ONE

Dear Dr. Zhang, 

I'm pleased to inform you that your manuscript has been deemed suitable for publication in PLOS ONE. Congratulations! Your manuscript is now being handed over to our production team.

Kind regards, 

on behalf of

Dr. Paolo Aurello 

Academic Editor

PLOS ONE